# Patterning of Polymer-Functionalized Nanoparticles with Varied Surface Mobilities of Polymers

**DOI:** 10.3390/ma16031254

**Published:** 2023-02-01

**Authors:** Shuting Gong, Tianyi Wang, Jiaping Lin, Liquan Wang

**Affiliations:** Shanghai Key Laboratory of Advanced Polymeric Materials, Key Laboratory for Ultrafine Materials of Ministry of Education, School of Materials Science and Engineering, East China University of Science and Technology, Shanghai 200237, China

**Keywords:** polymer-functionalized nanoparticle, polymer mobility, surface pattern, self-consistent field theory, broken symmetry

## Abstract

The polymers can be either dynamically tethered to or permanently grafted to the nanoparticle to produce polymer-functionalized nanoparticles. The surface mobility of polymer ligands with one end anchored to the nanoparticle can affect the surface pattern, but the effect remains unclear. Here, we addressed the influence of lateral polymer mobility on surface patterns by performing self-consistent field theory calculations on a modeled polymer-functionalized nanoparticle consisting of immobile and mobile brushes. The results show that except for the radius of nanoparticles and grafting density, the fraction of mobile brushes substantially influences the surface patterning of polymer-functionalized nanoparticles, including striped patterns and patchy patterns with various patches. The number of patches on a nanoparticle increases as the fraction of mobile brushes decreases, favored by the entropy of immobile brushes. Critically, we found that broken symmetry usually occurs in patchy nanoparticles, associated with the balance of enthalpic and entropic effects. The present work provides a fundamental understanding of the dependence of surface patterning on lateral polymer mobility. The work could also guide the preparation of diversified nanopatterns, especially for the asymmetric patchy nanoparticles, enabling the fundamental investigation of the interaction between polymer-functionalized nanoparticles.

## 1. Introduction

Phase separation in polymer-functionalized nanoparticles (PFNs) is a versatile strategy for patterning nanoparticles [1,2,3], generating nanoparticles with multiple patches or ripples on the surfaces [1,2,3,4,5,6,7]. Surface-patterned nanoparticles can act as colloidal analogs of atoms and molecules, behave as colloid surfactants, serve as model systems in studies of phase transition in liquid systems, and function as templates for the synthesis of hybrid particles [8,9,10]. Additionally, such nanoparticles can have a broad range of potential applications in fundamental research, diagnostics, colloidal stabilization, and so on [11,12]. Given their unique features and applications, the patterning of PFNs has been an active field for many experimental and theoretical investigations.

A significant advance in polymerization techniques allows the synthesis of PFNs with different dimensions, shapes, and chemical compositions [13,14]. In a typical synthesis, gold nanoparticles are first stabilized with cetyltrimethylammonium bromide or cetylpyridinium chloride and then replaced with thiol-terminated polystyrenes [1,15,16,17]. Several experiments indicate that the gold-thiolate interface is far from being static on the nanoparticles and show evidence of the mobility of thiolates on the surface of gold nanoparticles [18,19,20]. Both the diffusion and desorption of thiolates contribute to the dynamic feature of the interface. Murray and co-workers proposed that the diffusion of thiolates on gold nanoparticles is very slow [21], while other work indicated the fast diffusion of thiolates on gold nanoparticles [22]. Such contrasting results have two implications; that is, (1) the mobility of the polymer ligands attached to nanoparticles and (2) the dependence of the surface mobility of polymers on the external environment. For example, in the temperature-mediated patterning process—the number of patches per nanoparticle reduces upon the aging of PFNs at a higher temperature—the mobility of the polymer brushes can be enhanced as the temperature increases [23]. The surface mobility of polymer brushes is significantly impacted by preparation conditions such as solvent quality and temperature, usually leading to uncontrollable patterns. As such, physically understanding the effect of polymer mobility on the patterning of PFNs is particularly important for designing nanoparticles with specific patterns and functions. To date, however, there is not yet a clear picture concerning the role of polymer mobility in determining the surface patterns on nanoparticles. Investigation from the viewpoint of theory can help to gain insight into the role of polymer mobility on the surface patterns of PFNs.

Self-consistent field theory (SCFT) can be applied to clarify the effect of polymer mobility on the surface pattern of spherical nanoparticles. The SCFT, a polymer field theory implemented under a mean-field approximation, is a less computational yet reasonably accurate method for studying polymer systems [24,25,26,27,28,29,30,31,32,33,34]. One of the applications of the SCFT is to study the stabilization or phase separation of PFNs [35,36,37,38,39,40,41,42]. Roan et al. developed an SCFT and related numerical techniques for PFNs and found that the immiscible end-grafted polymers can confer multi-valence to nanoparticles [35]. Matsen et al. have implemented the SCFT to examine the effect of grafted brushes on the equilibrium distribution of PFNs in a block copolymer lamellar phase and revealed that the mixed brushes are more effective than the random copolymer brushes at positioning nanoparticles at the interface [43]. The SCFT was also developed to study the interaction between PFNs of various shapes and sizes [36,37]. In these works, the SCFT equations were solved in spherical coordinates, bi-spherical coordinates, or separate spherical–polar coordinates. Fredrickson et al. proposed a masking technique for polymer brushes, which allows the implementation of pseudo-spectral numerical SCFT methods. This unique technique can be easily extended to solve the SCFT on arbitrary geometrical substrates, for example, on a sphere [38,39,40,41]. Note that, in these studies, the polymers are treated to be either uniformly/randomly fixed or fully mobilized on the surface of PFNs. As such, few theoretical studies address the effect of polymer mobility on the patterning of PFNs, inspiring us to build a new SCFT model for PFNs with varied polymer mobilities.

We performed the SCFT calculation to examine the effect of polymer mobility on the surface patterns of a PFN. In this study, we do not intend to study polymer mobility through a specific dynamics simulation. Instead, we modeled the PFNs to contain both the immobile and mobile brushes on the nanoparticles. Here, we have assumed that the mobile polymers cannot desorb from the nanoparticle but can diffuse laterally on the surface. The mobility ability of the brushes depends on the fraction of mobile brushes. Using this model, we investigated the surface patterning of the PFNs under various conditions, including the fraction *f*_mb_ of mobile brushes, the particle size *R*_np_, and the grafting density *α*. The results show that the fraction of mobile brushes substantially influences the surface patterns of the PFNs, such as striped and patchy patterns. We found that for patchy patterns, the number of patches on a nanoparticle increases as the fraction of mobile brushes decreases. The work revealed that more patches are favored by the entropy of immobile brushes but are disfavored by the enthalpy associated with mobile brushes. We paid particular attention to the patchy nanoparticles and found that symmetry breaking usually occurs in the system. The present work could provide a strategy for preparing versatile nanopatterns, especially for the asymmetric patchy nanospheres, which enables the fundamental study of the interaction between PFNs.

## 2. Models and Methods

Simulations of PFNs were performed using self-consistent field theory (SCFT) [24,25,26,27,28,29,30,31,32,33,34]. An individual PFN is composed of *n*_b_ polymer brushes of *N* monomers and one nanoparticle. One of the polymer ends is dynamically or permanently tethered to the nanoparticle, where *n*_mb_ polymer brushes are mobile, and *n*_ib_ polymer brushes are immobile. Herein, the subscripts of mb and ib represent mobile brushes and immobile brushes, respectively. The volume fraction of mobile brushes is represented as *f*_mb_, and thus the number of mobile brushes is *n*_mb_ = *n*_b_*f*_mb_. The PFN is immersed in selective solvents. We assumed that no self-assembly happens between PFNs, and thus an individual PFN is chosen for the simulation. The nanoparticle is immobile in the center of the simulation box. The density $\phi_{\mathrm{np}}\left(r\right)$ of the nanoparticle is defined as follows. We first set the $\phi_{\mathrm{np}}\left(r\right)$ to be 1 as $\left|r-r_{\mathrm{np}}\right|\leq R_{\mathrm{np}}$ and then applied a so-called “cavity” field to obtain the smooth boundary condition of the nanoparticle [38,44]. Here, the **r**_np_ and *R*_np_ are the particle center and the radius of the nanoparticles, respectively. We treated the system as a semi-canonical ensemble: the number of PFNs is fixed, whereas the solvents (sol) are connected with a reservoir of the solution outside that maintains the chemical potential *μ*_sol_.

In the SCFT, one considers the configurations of a single PFN in a set of effective chemical potential fields $\omega_{k}\left(r\right)$, where k denotes ib, mb, and sol. We invoked incompressibility by introducing a Lagrange multiplier ξ(r). The free energy *F* of the system is [24]
(1)$$\begin{array}{l} \frac{F}{k_{B}T}=\frac{1}{v}\int dr\left\{\sum_{\begin{array}{l} i,j=\mathrm{ib},\mathrm{mb},\mathrm{np},\mathrm{sol}\\ \;i\neq j \end{array}}\chi_{\mathrm{ij}}\phi_{i}\left(r\right)\phi_{j}\left(r\right)-\sum_{k=\mathrm{ib},\mathrm{mb},\mathrm{sol}}\omega_{k}\left(r\right)\phi_{k}\left(r\right)+\xi \left(r\right)\left[\sum_{k=\mathrm{ib},\mathrm{mb},\mathrm{np},\mathrm{sol}}\phi_{k}\left(r\right)-1\right]\right\}\\ \;\;\;-n_{\mathrm{ib}}\ln Q_{\mathrm{ib}}-n_{\mathrm{mb}}\ln Q_{\mathrm{mb}}-z_{\mathrm{sol}}Q_{\mathrm{sol}} \end{array}$$
where *k*_B_ and *T* are Boltzmann constant and temperature, respectively. The Flory–Huggins parameter *χ* describes the local interaction between different components. $Q_{\mathrm{ib}}=\frac{1}{v}\int drq_{\mathrm{ib}}\left(r,N\right)$ and $Q_{\mathrm{mb}}=\frac{1}{v}\int drq_{\mathrm{mb}}\left(r,N\right)$ are the partition function of a single brush in the effective chemical potential fields $\omega_{\mathrm{ib}}\left(r\right)$ and $\omega_{\mathrm{mb}}\left(r\right)$, respectively. $q_{\mathrm{ib}}\left(r,s\right)$ (or $q_{\mathrm{mb}}\left(r,s\right)$) is a propagator for an immobile (or mobile) brush of length *s* that originates at grafted points. The propagators are determined by following the diffusion equation:(2)$$\frac{\partial q_{k}\left(r,s\right)}{\partial s}=\left[R_{g}^{2}\nabla^{2}-N\omega_{k}\left(r\right)\right]q\left(r,s\right)$$
where *R*_g_ is the radius of gyration of the polymers. For the mobile brushes, it is solved using the initial condition:(3)$$q_{\mathrm{mb}}\left(r,0\right)=\delta \left(\left|r-r_{\mathrm{np}}\right|-R_{\mathrm{np}}\right)$$

For the immobile brushes, the initial condition is defined by:(4)$$q_{\mathrm{ib}}\left(r,0\right)=\sigma_{\mathrm{ib}}\delta \left(\left|r-r_{\mathrm{np}}\right|-R_{\mathrm{np}}\right)/q_{\mathrm{ib}}^{f}\left(r,N\right)$$
where $q_{\mathrm{ib}}^{f}\left(r,s\right)$ is a complementary propagator that initiates from the free ends of the grafted chain. The propagator $q_{\mathrm{ib}}^{f}\left(r,s\right)$, subject to the initial condition, $q_{\mathrm{ib}}^{f}\left(r,0\right)=1$, satisfies the diffusion Equation (2). Similarly, the complementary propagator $q_{\mathrm{mb}}^{f}\left(r,s\right)$ of $q_{\mathrm{mb}}\left(r,s\right)$ also satisfies Equation (2) with the initial condition $q_{\mathrm{mb}}^{f}\left(r,0\right)=1$. $Q_{\mathrm{sol}}=\frac{1}{v}\int dr\exp\left[-\omega_{\mathrm{sol}}\left(r\right)\right]$ is the single-particle partition function of solvent in the potential field $\omega_{\mathrm{sol}}\left(r\right)$. The density profile $\phi_{k}\left(r\right)$ and the potential fields $\omega_{k}\left(r\right)$ are determined by the following self-consistent equations:(5)$$\phi_{\mathrm{mb}}\left(r\right)=\frac{n_{\mathrm{mb}}}{Q_{\mathrm{mb}}}\int_{0}^{N}dsq_{\mathrm{mb}}\left(r,s\right)q_{\mathrm{mb}}^{f}\left(r,N-s\right)$$
(6)$$\phi_{\mathrm{ib}}\left(r\right)=\frac{n_{\mathrm{ib}}}{Q_{\mathrm{ib}}}\int_{0}^{N}dsq_{\mathrm{ib}}\left(r,s\right)q_{\mathrm{ib}}^{f}\left(r,N-s\right)$$
(7)$$\phi_{\mathrm{sol}}\left(r\right)=z_{\mathrm{sol}}\exp\left[-\omega_{\mathrm{sol}}\left(r\right)\right]$$
(8)$$\omega_{\mathrm{mb}}\left(r\right)=\chi_{\mathrm{mb}-\mathrm{ib}}\phi_{\mathrm{ib}}\left(r\right)+\chi_{\mathrm{mb}-\mathrm{np}}\phi_{\mathrm{np}}\left(r\right)+\chi_{\mathrm{mb}-\mathrm{sol}}\phi_{\mathrm{sol}}\left(r\right)+\xi \left(r\right)$$
(9)$$\omega_{\mathrm{ib}}\left(r\right)=\chi_{\mathrm{mb}-\mathrm{ip}}\phi_{\mathrm{mp}}\left(r\right)+\chi_{\mathrm{ib}-\mathrm{np}}\phi_{\mathrm{np}}\left(r\right)+\chi_{\mathrm{ib}-\mathrm{sol}}\phi_{\mathrm{sol}}\left(r\right)+\xi \left(r\right)$$
(10)$$\omega_{\mathrm{sol}}\left(r\right)=\chi_{\mathrm{mb}-\mathrm{sol}}\phi_{\mathrm{mp}}\left(r\right)+\chi_{\mathrm{np}-\mathrm{sol}}\phi_{\mathrm{np}}\left(r\right)+\chi_{\mathrm{ip}-\mathrm{sol}}\phi_{\mathrm{ip}}\left(r\right)+\xi \left(r\right)$$
(11)$$\phi_{\mathrm{mb}}\left(r\right)+\phi_{\mathrm{ib}}\left(r\right)+\phi_{\mathrm{np}}\left(r\right)+\phi_{\mathrm{sol}}\left(r\right)=1$$

In this work, the nanoparticle is assumed to be neutral to the other components; thus, the Flory–Huggins parameters are set as $\chi_{\mathrm{mb}-\mathrm{np}}=\chi_{\mathrm{ib}-\mathrm{np}}=\chi_{\mathrm{np}-\mathrm{sol}}=0$. In addition, the immobile and mobile brushes are of the same components. As such, we set the Flory–Huggins parameters between immobile and mobile brushes as $\chi_{\mathrm{ib}-\mathrm{mb}}=0$. The other Flory–Huggins parameters are $\chi_{\mathrm{ib}-\mathrm{sol}}=\chi_{\mathrm{mb}-\mathrm{sol}}=0.8$ and $\chi_{\mathrm{np}-\mathrm{sol}}=0$ to account for the hydrophobicity of brushes and the hydrophilicity of the nanoparticles, respectively.

For clarity, we provide the nomenclature in Table 1.

The algorithm proposed by Fredrickson et al. was used to obtain a self-consistent solution [45,46]. We solved the diffusion Equation (2) with the Baker–Hausdorff operator splitting formula proposed by Rasmussen et al. [47] and used a two-step Anderson mixing scheme to update potential and pressure fields. The initial solution is a homogeneous state with a fluctuation amplitude value of 10^−4^. We performed the SCFT calculation in three-dimension space with spatial resolutions smaller than 0.2*R*_g_. For polymers, the chain contour was discretized into 100 steps [48]. The simulation continues until the relative accuracy in the fields reaches 10^−6^ and the incompressibility condition is satisfied [49,50].

## 3. Results and Discussion

The work presents an SCFT study on the phase separation of polymer ligands on a nanoparticle. The polymers grafted to the nanoparticle with a radius of *R*_np_ are either immobile or mobile (see Figure 1a). Here, we assumed that the mobile polymers can diffuse laterally on the surface but cannot desorb from the nanosphere. This system can be the nanoparticle attached with two kinds of polymers, where one is dynamically tethered and the other is permanently grafted. The system can also be a nanoparticle adsorbed by polymer ligands with different mobilities associated with external conditions such as temperature and solvent quality [21,22,23], where we used the fraction *f*_mb_ of mobile polymer to model the mobility of the polymer brushes roughly. In this system, the mobile and immobile brushes are of the same type and neutral to the nanoparticle. The polymer brushes are hydrophobic, whereas the nanoparticles are hydrophilic. In the SCFT calculation, we placed a PFN in the center of the box with periodic boundary conditions. The box is filled with explicit solvents. The effect of the *f*_mb_ on the patterning of PFNs with various grafting densities and nanoparticle sizes was examined.

### 3.1. Patterning of PFNs

Figure 1b shows characteristic patterns formed by the phase separation of polymers tethered on nanoparticles. They can be categorized into two main types: patchy nanoparticles (**P**) and striped nanoparticles (**S**). The patchy nanoparticles include single-patch nanoparticles (**P_1_**), two-patch nanoparticles (**P_2_**), three-patch nanoparticles (**P_3_**), and so on. Note that the subscript n in the representation of **P_n_** denotes the number of patches on a nanoparticle, and **P_0_** is the nanoparticle homogeneously covered with polymer brushes. The number and geometry of the strips on the striped nanoparticles can also be varied. Since the present work focuses on patchy nanoparticles, only one kind of striped nanoparticle is shown in the figure.

We systemically investigated the patterning of PFNs under the influence of nanoparticle radius *R*_np_, the fraction *f*_mb_ of mobile brushes, and the grafting density *α*. The obtained morphologies were summarized into morphology diagrams to present the role of the nanoparticle radius *R*_np_, the fraction *f*_mb_ of mobile brushes, and the grafting density *α*. Figure 2 shows the morphological diagram in the space of the grafting density *α* versus the fraction *f*_mb_ of mobile brushes for nanoparticles with various radii. As shown, only the nanoparticles with fewer patches were observed for small nanoparticles. For example, only the **P_0_** and **P_1_** exist at a radius of 0.4*R*_g_, whereas the **P_0_**, **P_1_**, and **P_2_** appear at 0.6*R*_g_. One can see that the patchy nanoparticle appears at high grafting density. At small *R*_np_, the boundary between **P_0_** and patchy nanoparticles shifts towards a low *α* value as the *f*_mb_ increases (see Figure 2a–d). Additionally, the number of patches on a nanoparticle decreases with increasing the *f*_mb_ at a certain *α* (see Figure 2b–h). For example, at *α* = 0.4*R*_g_*^−^*^2^ and *R*_np_ = 1.6*R*_g_, a morphology transformation from **P_6_** → **P_4_** → **P_3_** → **P_2_** → **P_1_** appears as *f*_mb_ increases (see Figure 2g). The striped patterns were observed for the PFNs with a radius larger than 1.0*R*_g_ at high *α* and low *f*_mb_ (see Figure 2d–h).

As the nanoparticle radius increases, the surface pattern becomes more diversified, and the morphology diagram becomes more complicated because patchy nanoparticles with more patches are formed (see Figure 2e–h). We note that the reentrant phenomena happen as the α value increases. For example, the **P_0_** is formed at either low *α* or high *α* values; the transformation from **P_1_** to **P_2_** and reentrant to **P_1_** occurs with increasing *α* value at *f*_mb_ = 0.7. As such, most of the boundaries between two patchy nanoparticles first shift to high *f*_mb_ and then turn to low *f*_mb_, as the *α* increases. Unlike the under boundary for forming **P_0_**, the upper boundary for forming **P_0_** shifts towards high *α* values as the *f*_mb_ increases. The striped pattern appears below the upper boundary for forming **P_0_**.

### 3.2. Role of Polymer Mobility

For the PFN, the nanoparticle is hydrophilic, but the polymers are hydrophobic. As such, the polymers incline to aggregate on the nanoparticle surface to reduce unfavorable interactions between polymers and solvents. However, aggregation behaviors are different for immobile brushes and mobile brushes. The immobile brushes are permanently attached to the surface of the nanoparticle and, therefore, tend to aggregate into uniformly distributed patches to avoid the overstretching of the tethered chains that are far away from the patches. The number of patches from immobile brushes increases as the nanoparticle radius increases. Due to mobility, mobile brushes have a strong tendency to aggregate into a single patch to minimize the unfavorable interface between solvents and polymers effectively. The appearance of patchy patterns with various numbers of patches, which depend on *f*_mb_, is a principal balance between the entropic loss of immobile brushes and the enthalpic reduction of mobile brushes. As the *f*_mb_ increases, the patterns with fewer patches are formed, which can reduce the unfavorable enthalpy without losing too much entropy of immobile brushes.

To prove the above conjecture, we calculated the enthalpy *U* and entropic loss –*TS* of PFNs with various patches using a seeding method [24,33]. To gain information on mobile and immobile polymers straightforwardly, we chose the PFNs with either fully mobile brushes (*f*_mb_ = 1) or entirely immobile brushes (*f*_mb_ = 0). Figure 3 shows the variation of enthalpy and entropic loss as a function of patch numbers. As shown, for the mobile brushes, the enthalpy increases dramatically, and the entropic loss shows a slight increase as the number of patches increases, while for the immobile brushes, the entropic loss has a considerable decrease, but the enthalpy exhibits a slight increase with increasing the patch number. This result is consistent with our conjecture that the formation of patchy patterns with various numbers of patches is mainly governed by the entropy of immobile brushes and the enthalpy associated with mobile brushes.

The density distributions were examined to understand the stretching of brushes on a nanoparticle. Here, the one-patch (**P_1_**) pattern was used as an example. Figure 4a shows the density distributions of immobile brushes in the **P_1_** pattern. One can see that the density of immobile brushes exhibits a calabash-like distribution, but the density away from the patch is very low. The appearance of the calabash-like distribution is because the free ends of immobile brushes are located in the patch (Figure 4b), but the other ends are permanently attached to the nanoparticle (Figure 4c). The calabash-like distribution was not observed for the mobile brushes (Figure 4d) because the grafted ends can be moved into the patch and assume similar distribution as the free ends (Figure 4e,f). The calabash-like distribution implies that the immobile brushes away from the patch are stretched to adapt to the patchy structures. For the nanoparticles with more patches, the immobile brushes are assigned to the nearest patches. Similarly, the immobile brushes are more stretched than the mobile brushes to accommodate the nearest patches.

The reentrant phenomena are also associated with the balance of enthalpic and entropic effects. Here, we took the morphological transformation of **P_1_** → **P_2_** → **P_1_** as a function of *α* value (*f*_mb_ = 0.7) as an example. At lower grafting density, the **P_1_** is formed, which can reduce the interfacial energy effectively, but the entropic loss is high. As the grafting density increases, the entropy loss sustainedly increases, and the enthalpic decrease cannot complement the entropic loss. Consequently, the **P_1_** cannot keep, and the **P_2_** with two patches appears. At higher grafting density, the **P_1_** with a large patch (relative to the patch of **P_1_** formed at low *α*) reappears because the formation of a large patch can somewhat alleviate the chain overstretching and decrease the unfavorable interaction energy. We want to emphasize that the **P_0_** formed at low *α* is impossible because the uniform grafting cannot be realized on a nanoparticle as grafting density is low. Under this condition, the fluctuation of grafting distribution should be included, and patchy patterns could be expected. However, this is beyond the scope of the present work. Therefore, the reentrance of **P_0_**, that is, an observation of **P_0_** at both low *α* and high *α* values, cannot happen in the actual systems.

### 3.3. Broken Symmetry

We noted in Figure 1 that the patches on a nanoparticle could be different in size. For example, the **P_3_** has three patches with different sizes, and the **P_5_** bears a large patch and four small patches on a nanoparticle. This phenomenon can be referred to as broken symmetry. The broken symmetry includes not only the size asymmetry but also the breaking of spatial symmetry. Figure 5 shows various **P_2_** patterns obtained for the PFN with *R*_np_ = 1.8 *R*_g_ and *f*_mb_ = 0.7. For the **P_2_** obtained at *α* = 0.2, as shown in Figure 5a, the size of the two patches is different, but their positions are highly symmetric. As the *α* increases, the positions of the two patches become less symmetric, and spatial symmetry breaking appears (Figure 5b). As *α* = 0.50, the two patches for **P_2_** are highly asymmetric in space, and the spatial symmetry breaking becomes evident (Figure 5c). The appearance of size asymmetry may be enthalpically favored because forming large domains can reduce unfavorable interactions with solvents. However, this could increase the entropic losses due to the stretching of polymers always from patches. Such chain stretching can be alleviated by changing the position of patches. Consequently, spatial symmetry breaking emerges for some systems.

### 3.4. Comparison with Existing Experiments

Gold nanoparticles decorated with thiol-terminated ligands are one kind of the most widely studied PFNs [1,3,15,18,23]. Over the past 30 years, a broad range of studies has been devoted to understanding the dynamic nature of thiol-gold bonds [51,52]. The studies revealed that the surface diffusion of thiol-tethered ligands could be accelerated by increasing the temperature. However, the effect of polymer mobility on the surface patterning of PFNs is not well known. Until recently, Kunacheva et al. have examined the effect of the mobility of polymer ligands, by changing the temperatures, on the patchy surface morphologies of gold nanoparticles end-grafted with thiol-terminated polymers [23]. They found that the gold nanoparticle with two well-defined patches is preserved as the grafting of thiol groups to the nanoparticle surface is strong (i.e., at a temperature lower than 40 °C). At higher temperatures, such as 80 °C, the number of patches per nanoparticle decreases, and the one-patch nanoparticle appears due to the enhanced lateral mobility of the polymer ligands. This observation is similar to our finding that the number of patches on a nanoparticle decreases as the fraction of mobile brushes increases, for example, as shown in Figure 2g. Because the increase in the fraction of mobile brushes could correspond to increased lateral mobility, the experimental and theoretical findings are well consistent. In addition, we also found a symmetry-breaking phenomenon in the patchy nanoparticles. It is hard to capture such a phenomenon in the experiment due to the limitation of characterization methods. Our work provides information for experimentalists to investigate symmetry-breaking phenomena in nanoparticle patterning in future work.

The difference between the experiment and theoretical calculation occurs upon the aging of the PFN at 80 °C over four days. They found in the experiment that a large number of gold nanoparticles lost the polymer ligand, and the bare gold nanoparticle appeared due to the desorption of polymer ligands at the increased solvent quality. This phenomenon, however, does not occur in our calculation because the SCFT model does not include the desorption of polymer brushes. Since the calculated results can capture the general feature of surface patterning by comparing it with existing experimental observations, the SCFT model could be used to predict the ligand-mobility-dependent surface morphologies of PFNs with various geometries of nanoparticle cores and different compositions of polymer brushes.

Finally, we outlined future work that may be relevant to the PFN community. First, the present SCFT model does not include the desorption of polymer brushes. Developing SCFT, including the effect of the desorption–adsorption of polymer ligands, could be an interesting topic in the future. Second, the SCFT ignores the fluctuation influence [24] and, therefore, the present method may not apply to study PFNs with a low grafting density where the fluctuation is crucial. Adopting field theory simulation can help solve this problem [24]. Third, we only considered an isolated PFN in the present system. The association of PFNs usually happens at high concentrations, which is akin to polymerization but occurs at larger length scales [53]. Investigating the origin of the PFN association by calculating the potential of mean force could be the aim of future studies. Fourth, predicting the properties of PFNs could be significant for applications such as energy harvesting [54] and welding [55].

## 4. Conclusions

We modeled PFNs containing both immobile and mobile brushes in the SCFT framework and studied the effect of polymer mobility on the surface patterning of PFNs. The polymers are found to form stripe patterns and patchy patterns on the nanoparticle surface, which depend on the nanoparticle radius, the grafting density, and the fraction of mobile brushes. The number of patches in a patchy nanoparticle increases with decreasing the fraction of mobile brushes. The work revealed that the dependence of patch numbers on the fraction of mobile brushes is dominated by the entropy of immobile brushes and the enthalpy associated with mobile brushes. We found that the symmetry of the patchy nanoparticles is usually broken in space and size. The enthalpy is essential in determining the size asymmetry, while the conformation entropy affects the spatial asymmetry. Comparing existing SCFT studies on PFNs, this is the first example investigating the patterns of PFNs combining mobile and immobile brushes and revealing the role of the immobile brushes. Since the SCFT ignores the fluctuation effect, the present method is not appropriate for studying PFNs with low grafting densities where the fluctuation may play an important role. Despite this, the work sheds light upon the influence of polymer mobility on the surface patterning of PFNs and could provide helpful information for preparing structured/functional nanoparticles with diversified patterns.

## Figures and Tables

**Figure 1 materials-16-01254-f001:**
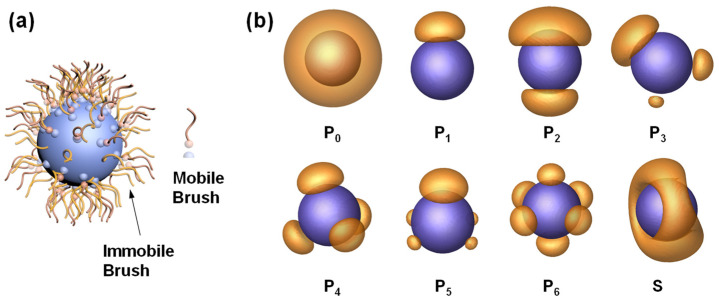
(**a**) Schematic illustration of a PFN consisting of mobile and immobile brushes. (**b**) Representative patterns formed by phase separation of polymers tethered on nanoparticles. **P_n_** denotes the n-patch nanoparticle; that is, **P_0_** is a zero-patch nanoparticle, **P_1_** is a one-patch nanoparticle, and so forth. **S** denotes the striped nanoparticle.

**Figure 2 materials-16-01254-f002:**
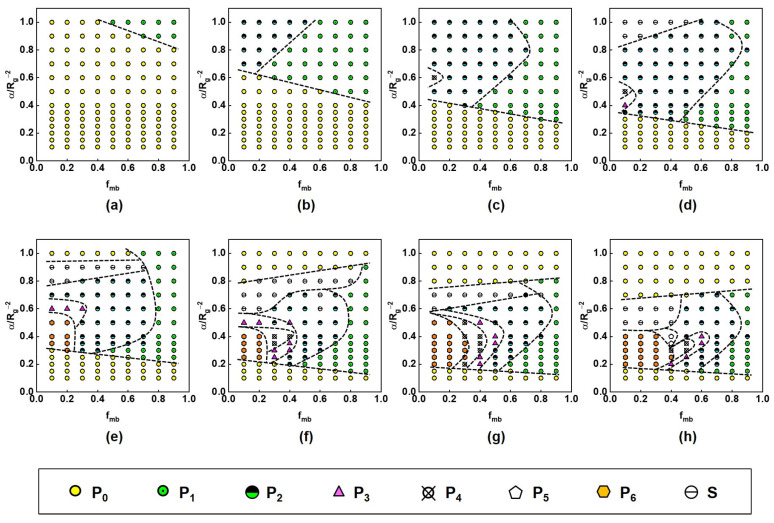
Morphological diagrams in the space of the grafting density *α* versus the fraction *f*_mb_ of mobile brushes for nanoparticles with the radii of (**a**) 0.4*R*_g_, (**b**) 0.6*R*_g_, (**c**) 0.8*R*_g_, (**d**) 1.0*R*_g_, (**e**) 1.2*R*_g_, (**f**) 1.4*R*_g_, (**g**) 1.6*R*_g_, and (**h**) 1.8*R*_g_.

**Figure 3 materials-16-01254-f003:**
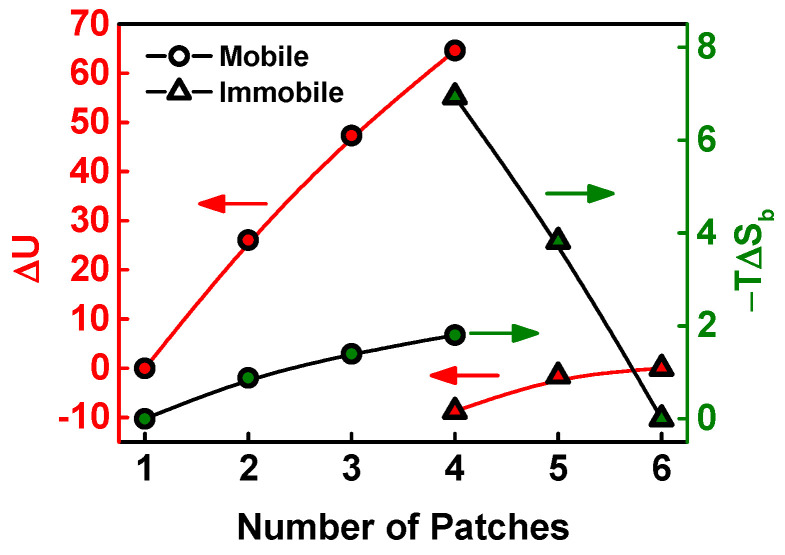
Variation of the enthalpy *U* and entropic loss −*TS*_b_ of brushes as a function of the number of patches on a nanoparticle. For the mobile brushes, the Δ*U* (and −*T*Δ*S*_b_) is the difference in enthalpy (and entropic loss) between the studied patchy patterns and the **P_1_** pattern, while for the immobile brushes, the Δ*U* (and −*T*Δ*S*_b_) is the difference in enthalpy (and entropic loss) between the studied patchy patterns and the **P_6_** pattern. The parameters are *R*_np_ = 1.8*R*_g_ and *α* = 0.4*R*_g_^−2^.

**Figure 4 materials-16-01254-f004:**
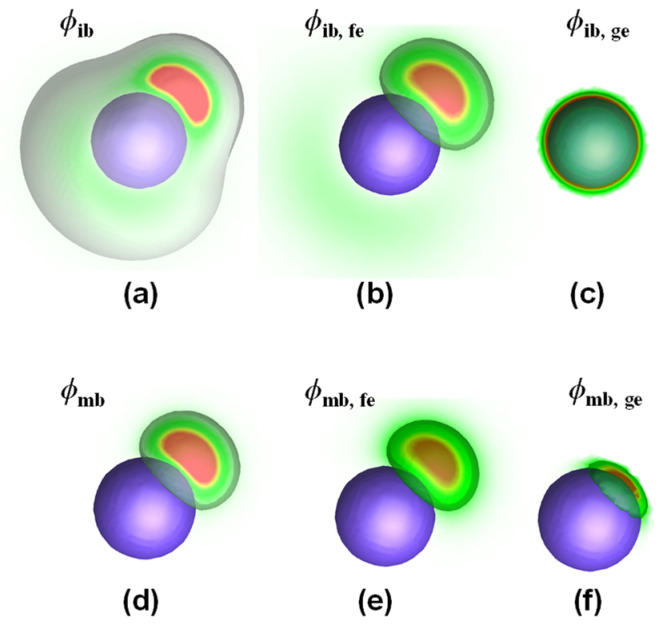
The density distribution of (**a**) immobile brushes (*ϕ*_ib_), (**b**) free end of immobile brushes (*ϕ*_ib,fe_), (**c**) grafted end of immobile brushes (*ϕ*_ib,ge_), (**d**) mobile brushes (*ϕ*_mb_), (**e**) free end of mobile brushes (*ϕ*_mb,fe_), and (**f**) grafted end of mobile brushes (*ϕ*_mb,ge_) in a single-patch nanoparticle (**P_1_**). The blue sphere denotes the nanoparticle, and the colors range from green (low density) to red (high density) in the density distribution.

**Figure 5 materials-16-01254-f005:**
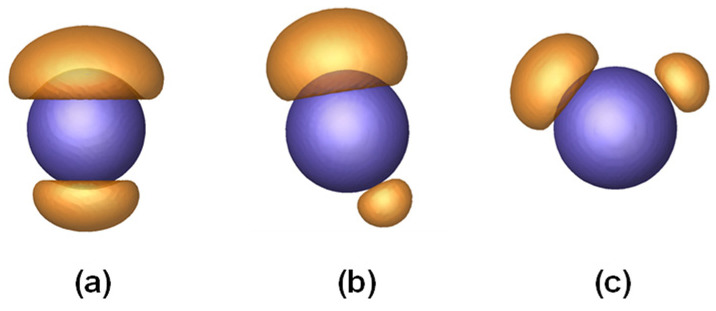
Two-patch patterns formed by phase separation of polymers tethered on a nanoparticle. The grafting densities are (**a**) 0.50*R*_g_^−2^, (**b**) 0.35*R*_g_^−2^, and (**c**) 0.20*R*_g_^−2^, respectively. The radius *R*_np_ of the nanoparticle is 1.8*R*_g_, and the fraction *f*_mb_ of mobile brushes is 0.7.

**Table 1 materials-16-01254-t001:** Nomenclature in the studies.

Symbol	Description
PFN	Polymer-functionalized nanoparticle
SCFT	Self-consistent field theory
mb	Mobile brushes
ib	Immobile brushes
sol	Solvents
*n* _b_	Number of polymer brushes on a nanoparticle
*n* _mb_	Number of mobile brushes on a nanoparticle
*N* _ib_	Number of immobile brushes on a nanoparticle
*N*	Number of monomers of a chain
*f*	Volume fraction of mobile brushes
$\phi_{\mathrm{np}}\left(r\right)$	Density of nanoparticle at position **r**
**r** _np_	Position of particle center
*R* _np_	Radius of nanoparticles
*μ* _sol_	Chemical potential of solvents
$\omega_{k}\left(r\right)$	Potential fields, where k denotes ib, mb, and sol.
$\phi_{k}\left(r\right)$	Density fields, where k denotes ib, mb, and sol.
ξ(r)	Pressure fields, Lagrange multiplier
*F*	Free energy
*k* _B_	Boltzmann constant
*T*	Temperature
*χ*	Flory–Huggins parameter
*Q* _k_	Partition function of a single brush, where k denotes ib, mb, and sol
*q*_k_(**r**,*s*)	Propagator at **r** for s monomer, where k = ib, mb
*q_k_*^f^(**r**,*s*)	Complementary propagator at **r** for s monomer, where k = ib, mb

## Data Availability

Not applicable.

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
