# Peer review of "Patterning of Polymer-Functionalized Nanoparticles with Varied Surface Mobilities of Polymers"

_materials, 2023, doi:10.3390/ma16031254_

Round 1

Reviewer 1 Report

The present article entitled “Patterning of Polymer-Functionalized Nanoparticles with Varied Surface Mobilities of Polymers”. Thus, this reviewer recommends the publication of this work after addressing the following comments.

The abstract should provide some quantitative information.

The results and discussion sections should provide some important citations.

In page 4; lines 138-147 showed plagiarism. The author should be revised the sentences.

The conclusion should be revised to show the outstanding point of this work.

Typographical errors and superfluous spaces throughout the manuscript should be corrected.

Reviewer 2 Report

Dear Authors

Your designed study about polymer brushes tethered/untethered on nanoparticles is interesting for researchers. The selected parameters for study of effective parameters are proper. I recommend to publish the manuscript after:

1-Correct some errors in text such as in line 70 and 71 (reference error: 38-41).

2-Moderate correction must be done about grammatical and spalling errors.

Sincerely

Reviewer 3 Report

The authors investigated addressed the influence of lateral polymer mobility on surface patterns by performing self-consistent field theory calculations on a modeled polymer-functionalized nanoparticle consisting of immobile and mobile brushes. The results show that except for the radius of nanoparticles and grafting density, the fraction of mobile brushes substantially influences the surface patterning of polymer-functionalized nanoparticles. The striped patterns and patchy patterns with various numbers of patches can be formed under different conditions. We found that broken symmetry usually occurs in patchy nanoparticles, associated with the balance of enthalpic and entropic effects.

The present work can provide a fundamental understanding of the dependence of surface patterning on lateral polymer mobility and could provide a strategy for preparing diversified nanopatterns.

The paper will be ready for publication after major revision.

The authors need to interpret the meanings of the variables.

All equations need references, is applicable.

Add nomenclature section.

What are the main advantages of the studies 51 and 52?

Please highlight your contributions in introduction.

The paper is well-written, I have to thank you to your effort.

What are the main features in Figure 2?

The introduction should be supported by recent publication from mdpi that discusses the recent advances of polymeric materias and their applications:

Bistable morphing composites for energy-harvesting applications

Predicting characteristics of dissimilar laser welded polymeric joints using a multi-layer perceptrons model coupled with archimedes optimizer

The abstract should be rewritten to reflect the significance of the proposed work. The current abstract shows a lot of background information.

Conclusion: What are the advantages and disadvantages of this study compared to the existing studies in this area?

The inspiration of your work must further be highlighted.

Add future works as bullets.

The space between value and units may be eliminated.

The numbers for the all equations have to be provided.

Looking and wishes for the revised version.

Round 2

Reviewer 3 Report

Accept in present form.